# Photoactive and Luminescent Transition Metal Complexes as Anticancer Agents: A Guiding Light in the Search for New and Improved Cancer Treatments

**DOI:** 10.3390/biomedicines10030578

**Published:** 2022-03-01

**Authors:** Brondwyn S. McGhie, Janice R. Aldrich-Wright

**Affiliations:** Nanoscale Organisation and Dynamics Group, School of Science, Western Sydney University, Locked Bag 1797 Penrith South DC, Sydney, NSW 2751, Australia; b.mcghie@westernsydney.edu.au

**Keywords:** cancer, luminescent, chemotherapeutics, metal, iridium, ruthenium, platinum, osmium, rhenium, photodynamic therapy, photoactivated chemotherapy

## Abstract

Cancer continues to be responsible for the deaths of more than 9 million people worldwide each year. Current treatment options are diverse, but low success rates, particularly for those with late-stage cancers, continue to be a problem for clinicians and their patients. The effort by researchers globally to find alternative treatment options is ongoing. In the present study, we focused on innovations in inorganic anticancer therapies, specifically those with photoactive and luminescent properties. Transition metals offer distinct advantages compared to wholly organic compounds in both chemotherapeutics and luminescence properties. Here we report on the characteristics that result from discrete structural changes that have been expertly used to fine-tune their properties, and how diverse inherent luminescent properties have been widely employed to monitor cellular localization to photodynamic therapy.

## 1. Introduction

There has been fascination with the investigation of metal complexes for medicinal uses, including anticancer therapies, since successful clinical trials and approval of cisplatin in the 1970s, the first platinum(II)-based medicinal complex [1]. Platinum(II)-based chemotherapies continue to be important in worldwide clinical cancer treatment; they are prescribed in more than 50% of all current chemotherapy regimens. Although extensively used, it is well recognized that therapeutic improvements need to be made. Current research focuses on reducing their systemic toxicity and drug resistance by exploiting alternative mechanisms of action, as well as introducing targeting moieties or selectivity for cancer cells [2]. Metal-based photoactive and luminescent complexes have a key role to play in this research, including the development of novel mechanisms of action and unique targeting strategies; they also provide insights into the mechanism of action and how targeting is realized.

Metals offer advantages to therapeutics and luminescence, with synthetically available and diverse structures, and oxidation states that are readily tunable, allowing pharmacokinetics to be modulated without impairing the therapeutic effect [1]. Unlike organic compounds, inorganic complexes can be deployed as a stable inactive prodrug in the system, until they are activated in situ by the dissociation of labile ligands or activation by an external stimulus such as light. Transition metals offer long fluorescent lifetimes, photostability, and long Stoke shifts, enabling them to outcompete autofluorescence, withstand irradiation for extended periods, and avoid self-quenching [3]. Strong luminescence properties provide research and clinical advantages, from benchtop binding studies, in vitro localization and analysis studies, to in vitro distribution and activation potential. Luminescent complexes can be used as probes to assess binding to biologically relevant molecules, such as DNA, or to monitor uptake, distribution, and the cellular microenvironment. Studying cellular localization and monitoring changes in the microenvironment can assist researchers to elucidate the mechanism of action of complexes, and allow the assessment of the suitability of promising complexes for use as chemotherapeutics before moving to in vivo studies [4,5,6]. Luminescence can also be used for bioimaging. It can be combined with an anticancer complex to afford a “theragnostic agent” that combines therapeutics and diagnostics agents in one molecule with one pharmacokinetic profile [3,7]. In addition to the passive advantages of luminescent anticancer complexes, some complexes utilize light-mediated pathways to induce their chemotherapeutic effect. These include photodynamic therapy (PDT), which utilizes photosensitizers, light, and oxygen to damage or kill cells, and photoactivated chemotherapy (PACT), where a prodrug is activated by light to release its chemotherapeutic capacity [8,9,10,11]. Luminescent complexes have strategic advantages in both research and clinical use when photophysical properties enable activation through the optic window of tissue. The optic window of tissue refers to the ability of different wavelengths of light to penetrate tissue (Figure 1). These properties have been investigated with the optical window, and therapeutic window ranges have been reported to be anywhere between 400–2000 nm. The therapeutically relevant window occurs between 600 and 950 nm [9,12,13,14,15]. Although the precise parameters of the optical window are contested, what is important is that researchers consider the activation wavelength when designing biological probes or photo-activated drugs [9,12,13,14,15]. To create an effective photo-activated drug, we must ensure that the irradiation wavelength matches the penetration depth of the tumor [16].

In this study, we aimed to provide an overview of transition metal complexes with anticancer and photoactive or luminescent properties that have been reported in the literature within the last five years. A systematic approach was used to identify potential publications, search strategies were developed for Scopus, Web of Science, and PubMed, and the search details are provided in the Appendix A. In simple terms, we searched for transition metal complexes with photophysical properties that were intended to exhibit anticancer activity. We focused not only on complexes that use their luminescence for clinical effect but also those with photoactive or luminescent properties that enable and facilitate the investigation of biophysical properties in vitro or in vivo.

## 2. Iridium Complexes

Iridium complexes are ideal candidates for anticancer therapies, and many researchers predict them to be the next big metal in the field. Research groups have developed complexes with cytotoxicity better than cisplatin, including in cisplatin resistant cells. Iridium complexes are reported to prevent the formation of resistance, and can re-sensitize cells to traditional platinum therapies. In this study, we focused on the many iridium anticancer complexes that exhibit photoactivity. Iridium complexes tend to feature high quantum yields, good photostability, large Stoke shifts, and long lifetimes. Many Iridium complexes are capable of two-photon fluorescence, which can be advantageous when trying to avoid crossover with biological auto-fluorescence. Most of these complexes are cyclometallated, and many have analogous basic structures with two phenylpyridine ligands together with a third non-cyclometallated ligand, which is further functionalized (typically, these are functionalized bipyridine or phenanthroline derivatives as shown in Figures 2–4). Because the iridium complexes discussed here follow similar structures, we categorized them based on where they localize within cells, which is commonly investigated using confocal microscopy studies.

### 2.1. Mitochondrial Targeting 

Mitochondrial targeting has been achieved by a variety of complexes with both simple and complex structures; it is thought that the intrinsic nature of iridium complexes results in mitochondrial targeting [17]. These complexes typically induce cytotoxicity by reactive oxygen species (ROS) production within mitochondria, triggering apoptosis. *Bis*-cyclometallated complexes (Figure 2: IR1) have demonstrated improved cytotoxicity compared to cisplatin and excellent potency in cisplatin resistant cell lines [18]. Modification of the third *N–N* ligand allowed the pharmacokinetic properties to be fine-tuned; for example, ester groups could be added where the length is easily modified to alter lipophilicity; increasing the length of the ester group also positively correlated with cytotoxicity, most likely due to an increased cell uptake (Table 1: IR2) [19]. These complexes (Figure 2: IR2) were shown to cause cell death not only through ROS production but also through a loss of mitochondrial membrane potential (MMP) and ADP depletion [2]. A structurally different option, the inclusion of a bidentate phosphorus ligand, exhibited high two-photon absorption properties, and resulted in mitochondrial-dysfunction-mediated cell death (Figure 2: IR3) [20]. This series of complexes is particularly interesting, because the rotational bonds of the phosphorous moiety made them viscosity sensitive, allowing real-time monitoring of mitochondrial viscosity. Organelle viscosity reflects its condition and function; thus, monitoring mitochondrial changes in viscosity will indicate changes in its functionality. Mao’s group produced so-called “mixed ligand” complexes, which have been reported to demonstrate pH-sensitive emission [21]. The complexes included one cyclometallated ligand and a tridentate ligand, and exhibit substantial luminescence with high quantum yields (Figure 2: IR4). These cyclometallated complexes caused apoptosis through ROS production and induced caspase activation upon irradiation. Their relatively short activation wavelength (425 nm) means that these types of complexes will be limited to superficial tumors. Mao et al. also reported several iridium(III) *N*-heterocyclic carbine complexes that are similarly *bis*-cyclometallated, and had cytotoxicity values that are better than cisplatin (Figure 2: IR5). Mitochondrial accumulation and the cytotoxicity of these iridium(III) *N*-heterocyclic carbine complexes were doubled upon irradiation (450 and 630 nm) with LED light [22,23]. Mao et al. revealed a complex series of interacting pathways that induce cell death with ROS production within the mitochondria, leading to reduced mitochondrial membrane potential, as well as caspase activation, which resulted in early apoptosis.

Other mitochondrial-targeted complexes include functionalized attachments, which themselves impart additional properties to the iridium(III) complex. For example, Ouyang et al. published a series of fluorinated complexes that displayed superior luminescence properties, as well as cytotoxicity that was two orders of magnitude better than cisplatin (Figure 2: IR6). These functionalized complexes showed a strong capacity to penetrate cell membranes and selectivity localize in the mitochondria. The attached moieties can influence the overall potency, particularly if they also have anticancer properties; for example, functional napthalamide based moieties can be anticancerous in their own right, and typically target the nucleus. Researchers reported that, with minor adjustments, different fluorescent and targeting properties can be realized, such that both the mitochondria and lysosomes could be targeted. A good cytotoxic effect, up to five times better than that of cisplatin, as well as some complexes with antimetastatic properties, have been reported [24]. Aspirin (IR7a) and salicylic acid (IR7b) have also been coordinated to *bis*-cyclometallated iridium(III) complexes with excellent chemo-anti-inflammatory results (Figure 2: IR7) [25]. Initial testing was extremely positive, showing a prevention of colony formation, migration, metastasis, and angiogenesis, as well as 10-fold better cytotoxicity than cisplatin. Coordination of these anti-inflammatory moieties appeared to bring about a good synergistic effect, with the complexes exhibiting immunoregulatory and anticancer properties. Mitochondrial accumulation can be traced and quantified using these complexes’ fluorescent properties; ROS generation triggers dysfunction, which induces both caspase-dependent apoptosis and caspase-independent paraptosis. Valproic acid, which can induce the differentiation of transformed cells and cause growth arrest, has also been conjugated to good effect (Figure 2: IR8), showing mitochondrial accumulation and better cytotoxicity than cisplatin in all cell lines tested [26]. Other groups reported similar success with nitroxide [27], HiPIP [28], and coumarin [29] (Figure 2: IR9, IR10, and IR11, respectively)—all of which are capable of accumulating in the mitochondria and permit the intracellular tracking and monitoring of physiochemical mitochondrial changes, such as the loss of mitochondrial membrane potential. Liu’s group published several “half-sandwich” mitochondrial-targeted complexes that demonstrate excellent luminescent, anticancer, and antibacterial properties (Figure 2: IR12). These complexes reportedly accumulated in the mitochondria or lysosomes with only small changes in the overall structure of the complex. These complexes induced cytotoxicity by ROS generation and changing the mitochondrial membrane potential, which led to apoptosis [30,31].

### 2.2. Lysosomal Targeting 

Unlike mitochondrial targeting, lysosomal targeting is not intrinsic to most iridium(III) complexes. Although most “half-sandwich/piano stool” complexes have been reported to target lysosomes [17], ligand modification within families of complexes can result in both mitochondrial and lysosomal targeting [21,24,30,31,32]. Other recent examples include complexes that incorporate imine-*N*-heterocyclic carbine, which has shown localization in the lysosome using fluorescent co-localization studies and exhibited very good IC_50_ values (Table 1:IR13) [33]. Some accumulation has been seen in the nucleus, but researchers have confirmed that DNA is not the target that induces cell death. Another recent example reported that both ruthenium and iridium piano stool complexes demonstrated good cytotoxicity and bio-imaging capability, allowing the determination of their intracellular fate in the cytosol or lysosomes, respectively (Figure 3: IR13) [34]. Past piano stool complexes (both fluorescent and non-fluorescent) were summarized previously [35]. 

Complexes that target lysosome incorporate unique structures such as P^P chelating ligand together with the typical *bis*-cyclometalated structure. Liu et al. published a diverse range of these, which featured cytotoxicity over eight-fold better than cisplatin and demonstrated good photo-properties that enabled them to be tracked to the lysosome (Figure 3: IR14) [36]. Cell death was mediated by ROS damage to the lysosome, and there was some evidence of changes to the MMP. The addition of a benzimidazole moiety resulted in a pH-sensitive complex that has IC_50_ values similar to that of cisplatin in cancerous cell lines but, interestingly, almost no cytotoxicity when administered in the dark (Figure 3: IR15) [37]. Liu et al. demonstrated pH-sensitivity, ROS production, and emission in lysosomes that are enhanced in acidic conditions, which is beneficial, as cancer cells tend to be more acidic than normal cells. These complexes can prevent several cancerous processes, such as cell migration, metastasis, invasion, colony formation, and angiogenesis. A series of published triphenylamine conjugated complexes showed that the modification of ligands enabled researchers to fine-tune the lipophilicity of these complexes, and, while they all accumulated in the lysosome and have good IC_50_ values, only one was selective of cancerous cells over normal cells (Table 1: IR16) [38]. The resulting complexes (Figure 3: IR16) acquired good fluorescence properties that enabled the identification of lysosomal accumulation and appear to be dual-functional with both metastasis inhibition and lysosomal mediated cell death.

### 2.3. Other Targets

Some iridium(III) complexes target DNA in the nucleus similarly to traditional platinum therapies. A new substructure of an organo–iridium–albumin bioconjugate was recently reported as the first of its kind, and had a long phosphorescent lifetime and high emission quantum yield (Table 1: IR17) [39]. This new structure (Figure 4: IR17) accumulated in the nucleus, but its cytotoxicity appears to be due to ROS generation, as it has a high ROS quantum yield. Observed nuclear accumulation was unexpected, as albumin bioconjugates do not usually transport into the nucleus. Excitingly, they were non-toxic in normal cells even after irradiation, but were highly cytotoxic in cancerous cells. Irradiation at 465 nm increased cytotoxicity in cancer cell lines. 

Another group has published a series of neutral iridium(III) complexes containing a single S atom that accumulates in the nucleus (Figure 4: IR18); however, the NH counterparts reported in the same paper were even more promising: both were light-activated and had improved PDT properties [40]. Some iridium complexes have been shown to bind with DNA via intercalation; these complexes typically have IC_50_ values similar to cisplatin, which correlate to their binding affinity [41,42]. A group of six inert iridium(III) complexes also exhibited nucleic-acid-binding properties, but fluorescent studies show that they accumulate in the actin cortex rather than the nucleus (Figure 4: IR19) [43]. There are several recent examples of iridium(III) complexes that interact with the endoplasmic reticulum; a group primarily focused on organic light-emitting diodes (OLEDs) was “swayed to the dark side” (to inorganic chemistry) using iridium complexes conjugated to guanidine and thiourea ligands (Figure 4: IR20) [44]. These inorganic complexes showed more consistent IC_50_ values across cell lines compared to their organic counterpart. Other complexes are reported to trigger a Ca^2+^ response via the endoplasmic reticulum to exert their cytotoxic effect [45] or efficiently prevent the translation of proteins by binding to the endoplasmic reticulum [46]; impressive cytotoxicity was achieved in both cases (Figure 4: IR21).

Unique iridium(III) luminescent complexes also include dinuclear iridium(III) complexes reported by Liu et al,. some of which incorporate similar cyclometallated ligands that have been discussed above, producing photophysical properties that are dependent on the bridging linker length. Although they have shown extremely promising photophysical properties, their cytotoxicity did not correlate with their ROS generation quantum yield or photo-cleaving capability, so further research is needed (Figure 4: IR22) [47]. The conjugation of death-receptor-binding TRAIL-mimicking peptide (a tumor-necrosis-factor-related apoptosis) induced apoptosis-inducing forms of a cancer-selective metal-peptide hybrid, which induces cell death via an extrinsic pathway (Figure 4: IR23). Researchers were able to confirm death receptor binding, which provides evidence that these luminescent complexes are promising anticancer agents, as they could detect cancer cells and induce cell death by the activation of intra- and extracellular cell death pathways [48]. Combining the known anticancer drug camptothecin and a cyclometallated iridium(III)) complex into a micelle “decorated” with cancer-targeting folic acid created a multi-action prodrug (Figure 4: IR24). Once internalized GHS reacts with the disulfide bond linking the iridium complex and camptothecin, which are both then able to enact their therapeutic effect —camptothecin via chemotherapy and iridium(II) via PDT. With strong luminescent properties, this micelle complex (Table 1: IR24) is also suitable for bio-imaging [49].

## 3. Ruthenium Complexes

Ruthenium complexes are a favorite when it comes to alternatives to the “traditional” platinum-based chemotherapeutics. Ruthenium complexes commonly have large Stoke shifts, high chemical, and photochemical stability; are typically highly water-soluble; and are resistant to photobleaching. In 2017, TLD1433, a ruthenium complex created by the McFarland group, became the first transition-metal-based complex to enter human clinical trials for the PDT treatment of cancer; in 2018, it succeeded in its goals in the trial and was terminated early [50]. A phase-III clinical study of TDL1433 commenced in 2019, which is expected to take 2–3 years to complete [51]. TLD1433, like many Ru(II) complexes, is inspired by the complex [Ru(bpy)_3_]^2+^, but other structures have been suggested for luminescent anticancer agents. Alternative structures include the addition of cyclometallated ligands, so-called piano stool geometry, and derivatives with appended peptides or long-chain hydrophilic groups.

### 3.1. Tris(bipyridine)ruthenium(II)-Inspired Complexes

Tris(bipyridine)ruthenium(II) ([Ru(bpy)_3_]^2+^)-inspired complexes were reviewed in detail at the end of 2018 and included in an account of the early development of TDL1433 by McFarland and colleagues [50]. Here, we provide a brief overview of TDL1433, focus on innovations within the past 5 years, and refer readers to McFarland’s review for seminal details [50]. TDL1433 contains functionalized aromatic chromophores with a low-lying triplet intra-ligand state, which makes them ideal for PDT. It is highly photosensitive and has long lifetimes, as well as ^1^O_2_ generation in both normoxic and hypoxic cells, despite having weak absorption within the biological window. This complex was designed with non-muscle-invasive bladder cancer in mind. This was done in a multidisciplinary approach described as “lateral”, as opposed to creating novel photosensitizers to find a suitable tumor target or retro-designing tumor specific complexes based on tumor properties. TDL1433 was part of a family of complexes tested with a range of different photophysical properties (Figure 5: TDL1433). The strategy of making premeditated incremental changes to the structure of tris N^N Ru complexes is well established; for example, the Gasser group published two complexes with the structure [Ru(phen)_2_(7R,8R-dppz-)]^2+^, where R was either OH or OMe. Both complexes displayed intense phosphorescence, good ^1^O_2_ quantum yields, but different cellular accumulation, as well as toxicity. Different toxic effects were reported for each complex before and after irradiation—at 420 or 800 nm for monolayer complexes or spheroids, respectively—which illustrates how functionalization of the intercalating groups can be utilized to fine-tune PDT properties (Figure 5: RU1) [52]. Trends are often inconsistent between families, and can be misinterpreted. For example, the addition of a large bulky group close to the metal center was thought to increase photodissociation, leading to toxicity; however, when *bis*-heteroleptic N^N complexes with ionizable groups, positioned to put a strain on the coordination sphere (Figure 5:RU2), were investigated in acidic and basic conditions, researchers showed that the deprotonation reduced dissociation and toxicity. Deprotonated complexes exhibited increased ROS quantum yields compared to their protonated analogs, suggesting that it is ROS and not photodissociation that gives the complex phototoxicity [53]. Systematic investigations of the effects of π conjugation to coordinated ligands in a series of heteroleptic ruthenium(II) complexes (Figure 5: RU3) demonstrated their theragnostic and tracking potential, both before and after photo treatment complexes, resulting in the largest ligands, with most π conjugation exhibiting the best PDT profile; there was also selectivity and a wide optic window, including near-infrared (NIR) light (Table 2: RU3), although blue light resulted in better IC_50_ values [54]. Estalayo-Adrián et al. recently published an investigation into the effect of long alky chains on [Ru(PHEN)_3_]^2+^ and [Ru(TAP)_2_(PHEN)]^2+^ complexes (where TAP = 1,4,5,8-tetraazaphenanthrene) (Figure 5: RU4a/b). The results showed that carbon alkyl chains had little to no effect on photophysical properties; but once a length of 21 carbons was reached, complexes showed increased emission lifetimes and quantum yields, improved uptake, increased ^1^O_2_ quantum yield, and increased cytotoxicity. Although the toxicity of the TAP 21-alkyl-carbon-chain complex increased, it was much less photoselective than the parent complex (Table 2: RU4b). On the other hand, the 21-alkyl-chain PHEN complex showed a remarkable increase in cytotoxicity compared to its parent complex and had a photo selectivity index of 27 (Table 2: RU3a) [55].

### 3.2. Piano Stool Complexes

Several ruthenium half-sandwich/piano stool complexes reported alongside iridium complexes are discussed above [24,30,32]. This geometry is capable of several different modes of interaction, which was succinctly demonstrated by a group of neutral ruthenium(II) arene complexes, some of which were phototoxic and others photoactivate. Phototoxic complexes of this type interact via groove binding, whereas the photoactivated complex releases the arene ligand upon irradiation and the metal binds covalently to DNA external to the groove (Figure 6: RU5) [56]. More recently, several ruthenium(II) piano stool arene complexes were tagged with napthalamide, which targets DNA via ROS generation in the nucleus (Figure 3: RU6). This complex also contains a morpholine moiety for lysosome targeting. Confocal localization studies revealed both lysosomal and nucleus accumulation, leading to cytotoxicity; unfortunately, however, although the toxicity was increased upon irradiation of blue light, they were still toxic in the dark (Table 2: RU6) [57]. Miachin et al. developed two luminescent derivatives of their triple-negative breast-cancer-active ruthenium(II) piano stool complex; they reported that the addition of a fluorophore changed the biological characteristics of the resulting complexes (Figure 6: RU7). They found that the cytotoxicity was reduced from 2.72 ± 0.11 to 68.23 ± 8.2 in MDA-MB-231 cells, and ICPMS experiments found that the complexes also varied in their cell localization and uptake [58]. Although these findings justify the conclusion that adding a fluorophore can still be used to build a more complete picture of cellular accumulation and mechanism as reported, it is clear that developing a cytotoxic agent with inherent fluorescence would be more beneficial.

### 3.3. Cyclometallated/Diverse Coordination Sphere

Cyclometallated ruthenium(II) complexes were previously thought to be too toxic in the dark to be used as effective PDT agents; however, more recent studies have found that some are non-toxic until photoactivated. Cyclometallated complexes bearing thienyl rings have been reported to exhibit promising properties compared to their N^N counterparts; for example, the former are more lipophilic and have photophysical properties more suited to biological application (for example, more red-shifted excitation (closer to the biological window). While complexes with only small thienyl groups were cytotoxic in the dark, more extended π-conjugated ligands showed good phototherapeutic properties with little or no toxicity in the dark (Figure 7: RU8a/b) [59,60]. Ruthenium(II) complexes with the N^O oxyquionolate-based ligand show promise for use in PACT and PDT, showing red-shifted absorbance compared to N^N complexes; they are also well within the therapeutic window at 600–700 nm (Figure 7: RU9). These complexes owe their toxicity to ROS-mediated DNA damage and show selectivity for *E. coli*, but further fine-tuning is needed to improve their cancer selectivity, as they had poor photo-selectivity for cancerous cells [61].

### 3.4. Other

Large ruthenium(II)-containing complexes, such as hydrogels and macromolecular complexes, have also been explored for use as delivery strategies to increase cellular uptake and increase selectivity for cancer cells. The smallest of these, the dinuclear metallointercalators, have an enhanced cellular uptake compared to their monomer counterparts (Figure 8: RU10). Although these metallointercalators are reported to bind to DNA using benchtop experiments, in vitro confocal studies revealed that they accumulated in the mitochondria and lysosomes and not in the nucleus. Their phototoxicity was activated by blue light, and they are potent in resistant cell lines via ROS generation [62]. Hydrogels are synthesized by the addition of glutaraldehyde solution to albumin and subsequently conjugated ruthenium(II) complexes. These hydrogels have demonstrated high selectivity for cancer cells and strong luminescence in cell imaging, making them not only able to be used as a delivery strategy, but also as cytotoxic theragnostic agents (Figure 8: RU11) [63]. The conjugation of polymers to the photodynamic agent [Ru(BPY)_3_]^2+^ has been reported to increase cellular accumulation. Mascheroni et al. published two such complexes which, although not very cytotoxic, serve as a “proof of concept” for the use of biocompatible polymers to modulate charge and cellular accumulation (Figure 8: RU12) [64]. When terazol ligands were coordinated to ruthenium(II) the resulting complex (Figure 8: RU13) was capable of forming nanoparticles that showed exceptional cellular accumulation but negligible cytotoxicity in the dark even at high concentrations. These complexes inhibited tumor growth and migration after excitation at 490 nm, which is just outside the ideal optic window for tissue [65]. 

Functional moieties with specific anticancer or targeting properties can be used to further modify the photophysical and pharmacokinetics properties of metal complexes. The addition of long hydrophilic tails forms surfactant ruthenium(II) complexes, which are localized to preferentially bind to DNA via hydrophobic interaction (Figure 8: RU14). They demonstrate anticancer and antimicrobial properties which have been attributed to ROS generation. These surfactant complexes are one of only a few examples of fluorescent complexes used to monitor cell death rather than activating it [66]. Peptide localization is a commonly used targeting strategy in metal chemotherapeutics, but is a less common strategy for luminescent complexes where researchers rely more on light activation to produce selectivity. Zhao et al. have reported a ruthenium(II) complex (Figure 8: RU15) that is functionalized by cervical cancer targeting peptide, which undergoes ligand substitution, releasing the therapeutic drug after reaching the tumor microenvironment. This prodrug has one- and two-photon-induced luminescence for deep penetration and is intended to be used as a diagnosis tool. Although the cytotoxicity was approximately equivalent to cisplatin, with such high selectivity this prodrug is very promising [67]. A dual-action europium–ruthenium complex (Figure 8: RU16) is another promising prodrug that is unusual in that the linker group absorbs visible light, which is then transferred to either the ruthenium(II) or europium moieties. The linker can be irradiated at different wavelengths for different functionalization. Irradiation at 350 nm allowed evaluation of cellular accumulation, after which it could be irradiated at 488 nm to release the ruthenium(II) drug; it could also be monitored by irradiation with 700 nm, which activated the europium fluorescence (Table 2: RU16) [68].

Nitric oxide performs regulatory functions in the heart and nephrons but can also have a tumoricidal effect, or conversely induce tumor growth, depending on the timing, concentration, and location. Utilizing this property, ruthenium(II) complexes have been created that are conjugated to nitrile and fluorescent dye conjugates. These complexes localized in the mitochondria and produced NO as desired, which inhibited mitochondrial respiration, triggering apoptosis [69]. Cancerous cells are thought to survive through the extension of the telomeric region, making them a target for anticancer therapies. Photoreactive ruthenium(II) quadruplex-DNA-targeting complexes have been demonstrated to preferentially bind by end-on π–π stacking to quadruplex DNA in preference to B-DNA. These complexes generated ROS and were photoselective, making them promising PDT agents [70].

## 4. Platinum Complexes

Platinum complexes have been the preferred chemotherapeutic treatments since the discovery and approval of cisplatin in the 1970s. Established platinum(II) complexes, such as like cisplatin, induce side effects due to their low specificity, and both intrinsic and acquired resistance is experienced with many cancers. Despite these disadvantages, a platinum drug will likely be part of one out of every two clinical patients’ chemotherapy treatment. Recent research has focused on increasing the specificity of new complexes to lower side effects and ensuring they are not cross-resistant with cisplatin. Here we focus on luminescent platinum complexes that are opening new pathways to targeting and monitoring cancer cells. Unlike iridium and ruthenium, platinum itself is not innately luminescent; however, with the judicious choice of ligands, such as cyclometallation (sometimes called cycloplatination), or the conjugation to a fluorescent or photosensitizing moiety, options become available. 

### 4.1. Photosensitizers

Photosensitizers imbue complexes with PDT properties that trigger cell death via ROS-mediated damage. They are typically desired to be non-cytotoxic until they are irradiated, which allows the precise treatment of the tumor area without systemic toxic effects. For example, Zhong et al. reported the creation of a Y-shaped PDT complex with three platinum centers and a photosensitizer that forms a complex that exhibited a remarkable improvement in cytotoxicity when irradiated (it was less cytotoxic than cisplatin, however). These Y-shaped complexes (Figure 9: PT1) accumulated in the nucleus, and ROS generation triggered a DNA damage response that led to apoptosis [71]. Also accumulating in the nucleus, square planar platinum(II) complexes conjugated to weak photosensitizers are not cytotoxic in the dark, but became more cytotoxic than cisplatin after just 5 min of irradiation (Table 3: PT2). These complexes (Figure 9: PT2) produced ROS, although it is unclear whether cell death was the result of ROS damage alone or platinum covalent binding to DNA also contributed [72]. Complexes that release or produce chemotherapeutic platinum agents upon irradiation are called PACT agents, and can be incorporated into PDT. For example, a small family of tertpyridyl complexes have been reported to be dual-action drugs, because they produced ROS and an active chemotherapeutic agent that disrupted DNA replication by intercalation (Figure 9: PT3) [73]. In some instances, the chemotherapeutic agent was released via reduction; for example, Thiabbaud et al. reported platinum(IV) complexes with a conjugated texaphyrin (Figure 9: PT4), which was potent when reduced by irradiation, and also produced ROS. Interestingly, this type of complex formed more DNA adducts when exposed to light, rather than being reduced by a biological agent [74]. Some of these dual-action complexes (Figure 9: PT5) are highly fluorescent and indicate local O_2_ levels that have enabled researchers to activate (irradiate) the complex when O_2_ was abundant, therefore maximizing the photosensitizer effect and ensuring cancer localization [75]. Dinuclear-platinum(II)-diethienylcyclopentene-based complexes (Figure 9: PT6) comprise a unique example of PACT agents, where irradiation by light results in the ligand “opening” and “closing”. When “closed”, the complex becomes extremely cytotoxic, more so than cisplatin, and exhibits a higher binding affinity for DNA through intercalation (Table 4: PT6) [76]. Other PDT complexes that are theragnostic are both therapeutic and can assist in diagnostic imaging. Platinum(II)-chlorin-type theragnostic agents exhibit photoactivated toxicity via ROS-triggered mitochondrial damage, but this also enables the monitoring of biodistribution. This feature allowed researchers to maximize tumor growth inhibition by waiting for a high accumulation within the tumor site before irradiation (Figure 9: PT7) [77]. A “theragnostic O_2_” meter has been reported, which enables the monitoring of local O_2_ levels similar to that mentioned above (PT6); but in this case, ROS generation is the sole cytotoxic mechanism. These complexes showed real-time monitoring via their phosphorescence, and exhibit advantages over traditional PDT cancer targeting (Figure 9: PT8) [78].

Researchers have been required to investigate photosensitizers that can be activated by more deeply penetrating irradiation, as UV light barely penetrates tissue. Upconversion nanoparticles use a multiphoton process, where the sequential absorption of NIR photons results in the emission of a single photon with a higher optical density. This approach has been utilized to create nanoparticles that converted deep-penetrating NIR to the “UV zone”, allowing the platinum(IV) prodrug to be reduced to cisplatin. This nanomedicine was designed to release the prodrug (Figure 9: PT9) upon reaching the acidic environment of the cancer site, which was then irradiated to activate it (Table 3: PT9) [79]. The first four-photon absorption by a platinum complex with very promising anticancer activity was reported by Zhang et al. They targeted lysosomes, which increased selectivity, as cancer cells are more prone to lysosomal-mediated cell death (Figure 9: PT10) [80]. Another more recently published nanoparticle converted NIR to UV light. In this case, the nanoparticle was coated with photoactivatable platinum(IV) and SiRNA targeting photo-linkers to produce a multifunctional “polyprodrug” (Figure 9: PT11). This nanoparticle(PT11) was observable using MR, CT, and UCL imaging for guided NIR-activated chemotherapy [81]. A different approach is to utilize the deep penetration of NIR by converting NIR light to heat; this is called photothermal therapy. San and colleagues have published a strategy that incorporated platinum(II) metallacycles with a fluorescent dye into multifunctional melanin dots (Figure 9: PT12). They reported a high cytotoxicity, which was attributed to the synergistic effect of photothermal activation and the chemotherapeutic effect of the activated platinum complex [82]. PDT complexes that are activated with X-rays (X-PDT) have been proposed as a deep-penetrating option; Wang and colleagues have synthesized platinum(IV)-coated nanoparticles that upon X-ray irradiation released both ROS and cisplatin (Figure 9: PT13). The platinum(IV) acted as a sacrificial electron acceptor to increase the OH yield, resulting in an impressive synergistic nanoparticle [83].

### 4.2. Fluorescent Moiety

Fluorophores can be utilized to elucidate the biological fates of drugs and in turn their mechanisms of action. Familiar fluorophores include coumarin and thiazol derivatives, which have both been conjugated to platinum in the past with great success. More recently, coumarin and thiazol ligands have been coordinated with platinum and palladium complexes to produce cytotoxic complexes that could act as chemosensors in cancer cells (Figure 10: PT14), where their fluorescence indicated the presence of CN^−^ [84]. In other work, large coumarin and thiazol hybrids have been attached to platinum and palladium as both tetradentate and bidentate ligands, producing a family of complexes with diverse fluorescent and cytotoxic properties. It was reported that the metal and type of coordination had a significant impact on both cytotoxicity and fluorescence, with some showing significantly improved cytotoxicity compared to cisplatin. The palladium coumarin bidentate derivative had impressive fluorescence properties, which were detectable above the background cellular fluorescence, revealing that it was localized in the nucleus [85]. Imidazole phenanthroline not only brings fluorescence to the complexes, but imbues antitumor and antimicrobial properties; it has recently been coordinated with platinum and palladium together with 1,2-diaminocyclohexane (Figure 10: PT15). In this case, the resulting platinum complexes (Table 3: PT15) were more active than the palladium complex, a trend that was also evident in their DNA-binding affinity via intercalation; the inherent fluorescence properties could be used to probe for DNA binding [86]. A small family of polyamide conjugated complexes (Figure 9: PT16) has been created that incorporates multiple DNA-binding moieties, including minor and major groove binders as well as intercalating moieties. These complexes demonstrated moderate-to-good penetration of spheroids, which is promising for treating tumors some distance from the vasculature (Table 3: PT16) [87].

### 4.3. Cyclometallated Platinum Complexes

The organo–metallic bond in cyclometallated complexes results in a net deprotonation of the carbon, making it a strong σ-donor, whereas the polypyridyl group is a good π-acceptor; these properties together create a very strong ligand field. The increase in the energy gap between the unoccupied and occupied orbitals caused by the σ-bond results in luminance. Cyclometallation does not necessarily impart toxicity, but allows fluorescent tracking within cells, which can assist with intelligently fine-tuning the structure while determining the cellular localization and potentially the mechanism of action of the complex. Millan et al. produced neutral cyclometallated platinum(II) complexes (Figure 11: PT17) with good cytotoxicity (Table 3: PT17), and their fluorescence allowed the detection of the cellular internalization and localization in the cytoplasm [88]. Platinum(IV) cyclometallated complexes are less common, but a promising series of platinum(II and IV) complexes were published with cyclometallated tetradentate ligands (Figure 11: PT18). They were reported to exhibit approximately equivalent cytotoxicity to cisplatin, but were not cross-resistant with cisplatin, and had much higher selectivity and cellular accumulation. The fluorescent properties of these complexes could be fine-tuned by altering the size of the N^N chelating ring or the substitutes of the aryl ring [89].

## 5. Rhenium Complexes

Rhenium is a more recent addition to the arsenal of metals used for chemotherapeutics. Re(I) complexes based on the parent tricarbonyl [Re(I)(CO)_3_]^+^ complex have been of particular interest, owing to their distinguishing triplet-based luminescent emission, which can be used for fluorescent microscopy imaging applications and has the ability to release CO in situ, which adds to their cytotoxicity [90]. In fact, in our search, all examples of rhenium luminescent anticancer complexes were based on this [Re(I)(CO)_3_]^+^ parent complex. Research exploring the relationship between structure, luminescence, chemotherapeutics, and pharmacokinetics has been undertaken, which explored the effects of the charge, lipophilicity [91], and basicity of ligands [90]. Although the rhenium(I) complexes studied were not particularly cytotoxic, the positively charged complex was more cytotoxic than the non-charged, which was explained by the former’s increased lipophilicity [91]. The axial N^N ligand was reported to have negative correlations with luminescent quantum yields and energy, but no correlation with cytotoxicity was observed [92]. The addition of water-soluble phosphines (Figure 12: RE1) rather than polyaromatic axial N^N ligands produced better luminescence and cytotoxicity. The phosphine ligand regulated luminescence, which strongly correlated with cytotoxicity. These rhenium(I) complexes were phototoxic, having high photoselective indices and mechanisms of action rationalized to result from the release of CO and ROS production upon irradiation (Table 4: RE1) [93]. As mentioned earlier, rhenium(I) complexes have excellent luminescent properties for imaging, making them good candidates for theragnostic agents and monitoring of intracellular conditions. For example, the addition of dinitrophenyl sulfonylaminomethyl (DNPS) has created a GHS responsive bio-sensing unit, and this complex (Figure 12: RE2) has showed low cytotoxicity and weak luminescence due to DPNS quenching; however, cellular interaction with GHS resulted in DPNS leaving, which triggered emission and cytotoxic enhancement (Table 4: RE2) [94]. Mao’s group has worked on mitochondrial-targeted theragnostic rhenium(I) complexes (Figure 12: RE3), which altered the metabolism, and reported upon O_2_ consumption, where ROS generation and respiration inhibition ultimately led to mitochondrial-mediated apoptosis. These complexes offered real-time tracking of mitochondrial changes, and had superior cytotoxicity to cisplatin, in addition to selectively killing cancer cells when co-cultured with normal cells [95,96]. This group also published the first reported rhenium(I) complex to induce autophagy-mediated cell death (Figure 12: RE4). All of the analogous complexes conjugated to β-carboline derivatives that are cytotoxic in their own right, resulting in complexes with good cytotoxic properties and pH-dependent phosphorescence, which has been utilized for lysosomal imaging [97]. Not all [Re(I)(CO)_3_]^+^ complexes trigger normal modes of cell death, such as apoptosis, paraptosis, necrosis, and autophagy; for example, [Re(I)(CO)_3_ (H_2_O)]^+^ complexes (Figure 12: RE5) exhibited both good cytotoxicity and imaging luminescent properties, but induced a novel mode of cell death that was not triggered by ROS and does not fall into any of the categories previously mentioned [98]. Conjugation of a known ferroptosis-inducing agent, artesunate, produced a dual-mode rhenium(I) complex able to induce both apoptosis and ferroptosis. The resulting complex (Figure 12: RE6) demonstrated mitochondrial accumulation and high cytotoxicity towards cancer cells [99]. 

Dinuclear rhenium complexes, both homo- and heteronuclear, have also been explored, and show good potential as theragnostic agents. Heterodinuclear rhenium(I) β-carboline complexes (Figure 12: RE7) have given rise to good lysosomal targeting and luminescence, enabling the lysosomes in cancer cells to be visualized. They exhibited significant phototoxicity under blue-light irradiation, producing ROS that resulted in lysosomal damage. The rhenium(I) β-carboline complexes (RE7) had higher ^1^O_2_ yields under acidic conditions, which are expected to facilitate cancer targeting [100]. The gold(II)/rhenium(I) heterodinuclear complexes (Figure 12:RE8) that have also been produced benefit from the gold delivering increased cytotoxicity to the strongly fluorescent rhenium complexes. These complexes accumulated in the cytoplasm, localizing around the nucleus and inner edge of the cell membrane. They were photoselective: blue-light irradiation triggered up to a five-fold increase in cytotoxicity (Table 4: RE8) [101,102].

## 6. Osmium Complexes

Osmium complexes characteristically have low emission quantum yields and fluorescent lifetime; however, they are likely to have emission maxima in the near-infrared region, making them desirable luminescent complexes for biological applications [103]. Osmium complexes have been reported as anticancer agents, but have only more recently been evaluated as cellular imaging probes for a theragnostic approach to therapy. The key challenge for transition metal applications for biological luminescent agents is the limited number of complexes that can be excited within the deep-penetrating optic window of tissue; therefore, the propensity of osmium to be red-shifted makes them crucial to designing complexes that can achieve an enhanced depth of penetration to enable us to deliver anticancer agents to the interior of solid tumors.

Comparison with equivalent polypyridyl ruthenium(II) and osmium(II) complexes provides an excellent example of the red-shifting ability of osmium. While equivalent ruthenium(II) complexes excited at 459 nm produced emissions between 550 and 700 nm, osmium(II) complexes (Figure 13: OS1) can be excited up to 683 nm with emission between 700 and 850 nm. Osmium(II) complexes also exhibited different localization—interior accumulation in the lysosome—while ruthenium was observed in the mitochondria; both were highly photoselective and cancer-selective [104]. Other polypyridyl osmium(II) complexes accumulated in the mitochondria with NIR luminescence, although they were not photoactivated; instead, their intrinsic fluorescence was utilized to monitor their therapeutic effect [16]. Tridentate terpyridine complexes (Figure 13: OS2) have also produced NIR phosphorescent complexes that were photoselective and able to be activated with both red and blue light, with both wavelengths producing cytotoxicityies higher than cisplatin (Table 5: OS2) [105]. Half-sandwich osmium(II) complexes produced using sulfonamide and azopyridine ligands have also been explored. Only one sulfonamide complex (Figure 13: OS3) showed photoactivity, producing ROS to induce cell death, and red luminescence was utilized for ROS generation studies, uptake, and distribution studies [106]. The design objective was to be able to penetrate the tissue to the depth of a tumor with an effective phototoxic agent; Lazic et al. produced a “panchromatic” osmium(IV) complex (Figure 13: OS4), which could be activated with irradiation anywhere between 200 and 900 nm (Table 5: OS4). This series of osmium(IV) complexes had low toxicity in the dark and were effective in both hypoxic and normoxic conditions, making them a very exciting development in the field [107].

## 7. Other Metal Complexes

Other transition metals were identified in our searches which are worth mentioning. In this section, we describe the use of luminescence to assist in detecting the characteristics of these potential drugs, rather than focusing on how their luminescence can be used in the clinics. As transition metals, they deliver benefits in addition to the luminescence, such as long Stokes shifts (which prevent self-quenching, as seen in some organic fluorescent compounds) and increased potency for anticancer applications. The metals included in this section are zinc, cobalt, palladium, silver, and nickel. TOOKAD^®^ Soluble is the first and only transition metal PDT agent to be approved anywhere in the world. This palladium-based complex from Scherz for Steba Biotech completed phase-III trials in 2015 (ClinicalTrials.gov, Identifier: NCT01310894), and was later approved in some countries for localized prostate cancer. TOOKAD^®^ Soluble (Figure 14: TOOKAD^®^ Soluble) is derived from photosynthetic pigment bacteriochlorophyll α, which is used in nature to produce energy from sunlight, and is the improved version of prior work by the Scherz group [51,108].

Platinum(II) and palladium(II) complexes are sometimes published together, as they are isostructural, meaning they coordinate in the same square planar geometry. For example, the coumarin thiasol chemo-sensing ligands mentioned earlier were created as both platinum(II) and palladium(II) complexes (Figure 10: PT14). These fluorescent chemo-sensor ligands were able to detect F–, AcO–, and CN– ions while the metal centers conveyed cytotoxicity to the complex; in this instance, the palladium(II) complex was more cytotoxic than the platinum(II) [84]. The square planar geometry of these metals allowed them to be inserted into organic macrocycle rings; for example, nickel(II), platinum(II), and palladium(II) efficiently inserted into benzimidazole-based macrocyclic ligands (Figure 14: OM1). Palladium(II) complexes were phosphorescent, while the platinum(II) complexes were fluorescent; although the resulting properties have not been utilised, future applications for distribution studies or DNA interaction studies are probable. 

The nickel(II) complex (Figure 14: OM2) was not cytotoxic, although both the platinum(II) and palladium(II) complexes were considerably more cytotoxic than cisplatin, and have potential as both helical and quadruplex DNA binders [109]. Silver *bis*-methyl salen complexes have been investigated as potential anticancer agents and demonstrate promising fluorescent and DNA-binding properties. Their intrinsic fluorescence was used alongside other techniques to analyze DNA binding. These are promisingly non-toxic in normal cells, although assessment in cancer cell lines needs to be undertaken if they are intended to be anticancer agents [110]. Silver nanoparticles can also be fluorescent at larger physical sizes, but these are usually less cytotoxic than the smaller non-fluorescent form due to their larger size hindering cell uptake. To circumvent this, Fedorenko et al. developed silica nanoparticles containing terbium(II) for fluorescence, which were “decorated” with cytotoxic silver nanoparticles with amino groups for better internalization. The resulting complexes exhibited good cytotoxicity and were selective for cancer cells. Fluorescence was used to show that they were internalized and distributed within the cytoplasm; it has been suggested that they have a future as theragnostic agents [111]. Ma’s research team has created and published several zinc(II) complexes with fluorescent and anticancer properties. Six zinc hydroxyl-terpyridine complexes were reported to be more cytotoxic than cisplatin, and their fluorescence was utilised to show strong binding to hemoglobin (Figure 14: OS2) [112]. The halogen-substituted analog of these complexes was shown to elicit a more red-shifted fluorescence, which is promising for biological application, while keeping very good cytotoxicity. Fluorescence was used primarily for binding studies rather than in vivo or in vitro imaging, but these opportunities remain available for future investigation (Figure 14: OS2) [113]. Cobalt(II) complexes with the benzenesulfonamide ligand were synthesized and published with moderate IC_50_ values (32.02–42.5 μM); sulfonamides are typically antibacterial, but when bound to cobalt(II) they demonstrated increased fluorescence and anticancer potential [114].

## 8. Conclusions

The potential for theragnostic and luminescent therapies with metal complexes is extraordinary, as demonstrated by the innovations seen in the past five years summarized here. Many challenges have been overcome, such as increased photoselectivity and activation at deeply penetrating wavelengths well within the optical window. With TDL1433 currently in phase-III clinical trials, more confidence and interest are emerging in this field, so we can expect more innovations with dedicated resources. Understanding what makes these complexes successful in the clinic is key to uncovering new complexes tailored to specific cancers. There has been a significant deepening of our understanding of the mechanisms of action and structure-to-activity relationship, but future work should take note of the success of TDL1433 and start to consider specific cancer targets, as well as seeking research partners that can progress these innovative complexes towards the market.

We hope to have shone a light on the usefulness of luminescent complexes for both clinical applications and research tools, providing a comprehensive array of spectroscopic properties that facilitate the search for new chemotherapeutics and showing that these investigations are complementary. Additionally, throughout our literature search we noticed that many of us tend to devote our research to just one metal. We suggest that the field has an immeasurable opportunity for growth if groups (including our own) can diversify the metals we consider when designing new complexes; in doing so we could develop new insights into the role of metals in physiological and photophysical properties as well as the structure–activity relationship.

## Figures and Tables

**Figure 1 biomedicines-10-00578-f001:**
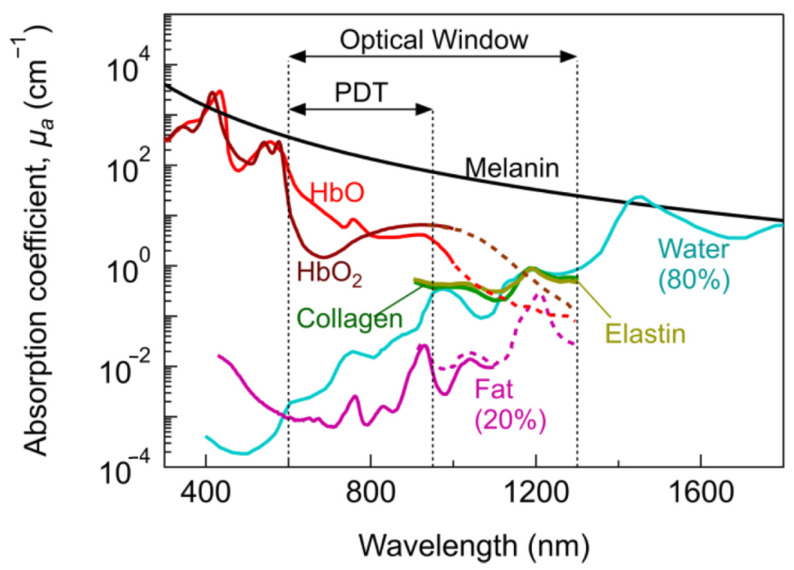
The therapeutic optic window; the absorption coefficient as a function of wavelength for several biologically relevant molecules, where HbO refers to hemoglobin, and HBO_2_ refers to oxyhemoglobin, as published by Algorri et al. [9].

**Figure 2 biomedicines-10-00578-f002:**
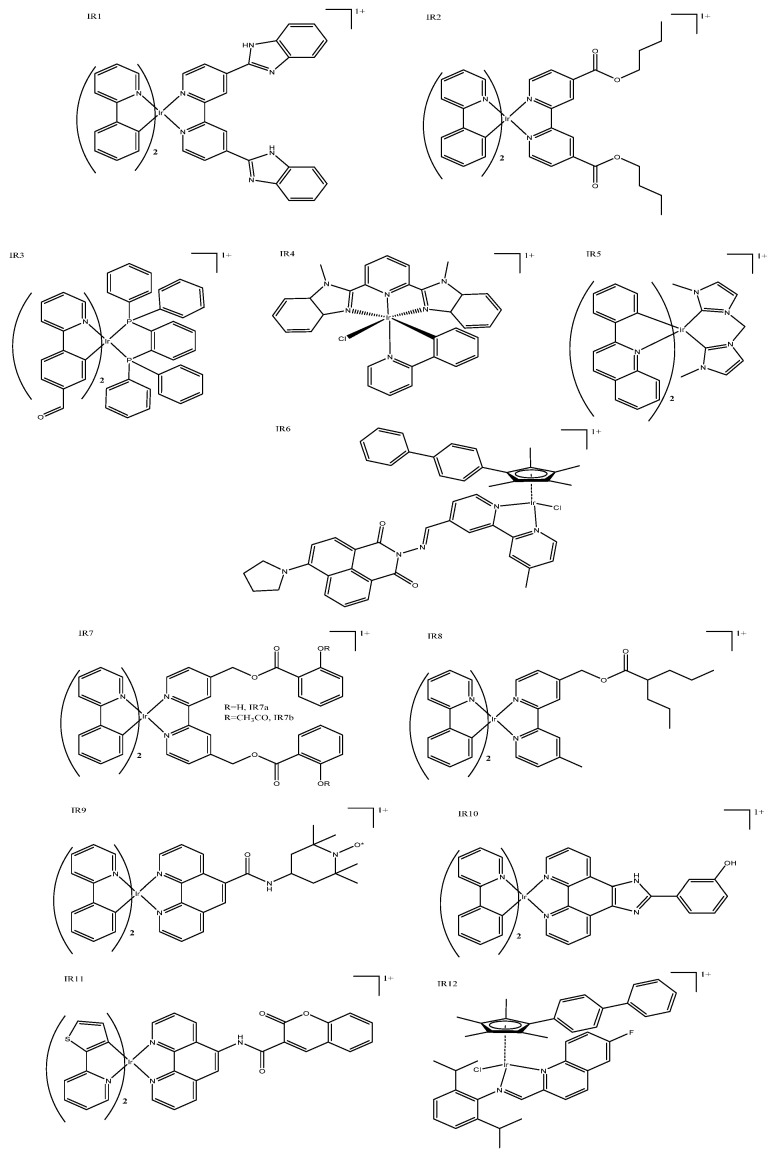
Mitochondria-targeting iridium complexes: cyclometallated complexes IR1 [18], IR2 [19], IR3 [20], IR5 [22], IR7a (where R = H (aspirin)), IR7b (where R = CH_3_CO (salicylic acid)) [25], IR8 [26], IR9 [27], IR10 [28], and IR11 [29]; half-sandwich complexes IR6 [24] and IR12, [30]; tridentate complex IR4 [21].

**Figure 3 biomedicines-10-00578-f003:**
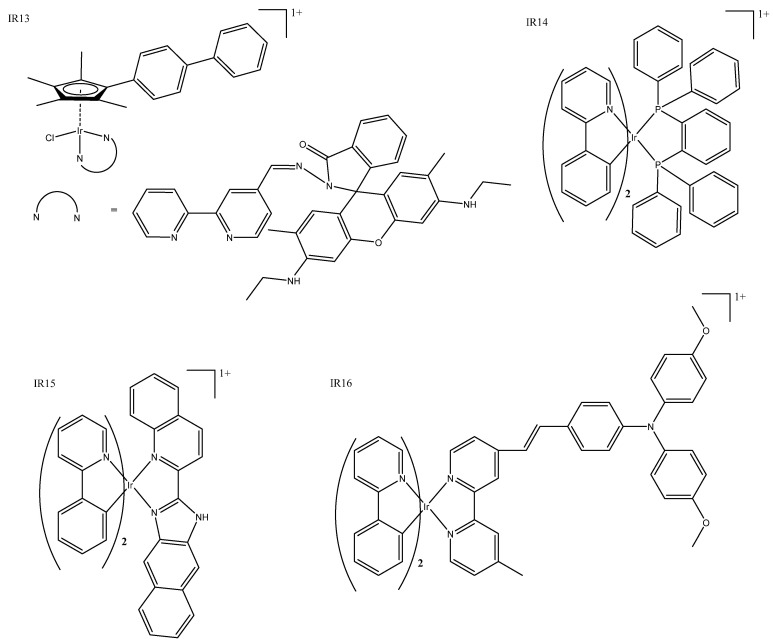
Iridium complexes: half-sandwich complex IR13 [34], cyclometallated complexes IR15 [37] and IR16 [38], and P^P complex IR14 [36].

**Figure 4 biomedicines-10-00578-f004:**
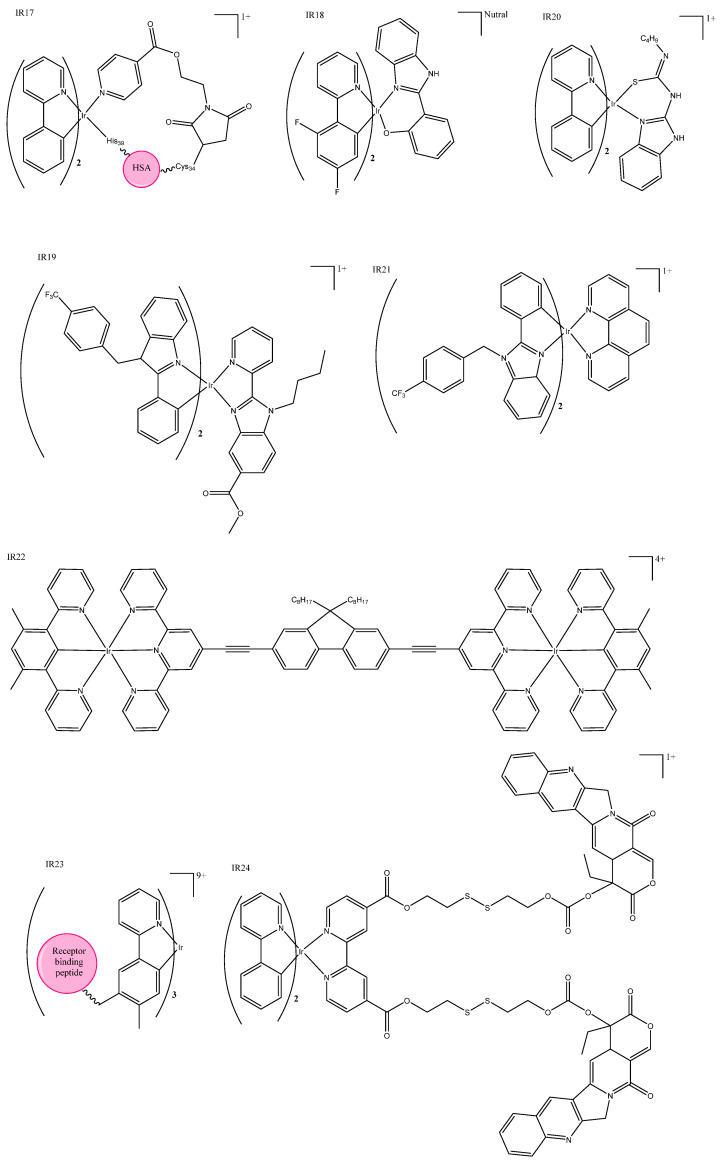
Iridium complexes: cyclometallated complexes IR17 [39], IR18 [40], IR19 [43], IR20 [45], and IR21 [46]; dinuclear iridium(III) complex IR22 [47]; TRAIL-mimicking peptide complex IR23 [48]; and micelle multi-action prodrug IR24 [49].

**Figure 5 biomedicines-10-00578-f005:**
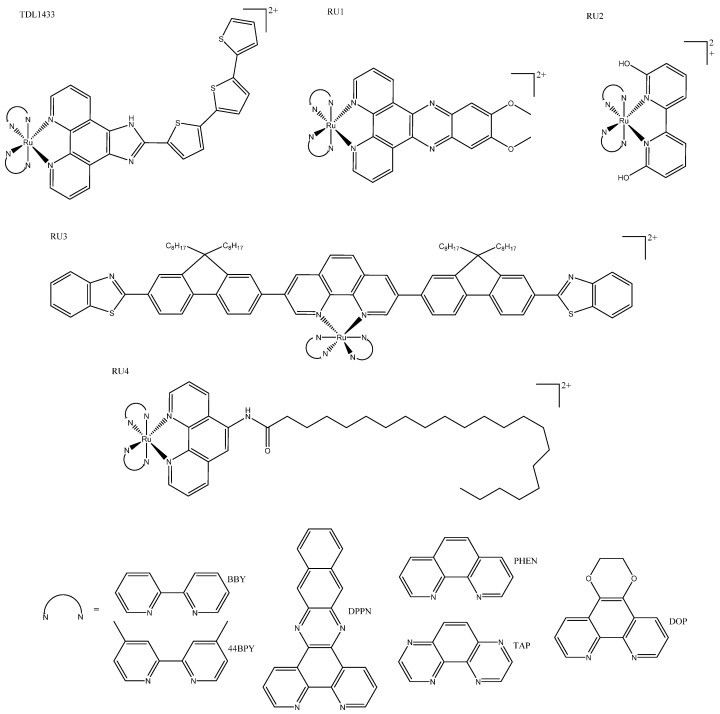
Tris(bipyridine)ruthenium(II)-inspired complexes: TDL1433, where N^N = 44BPY [50]; the similar complex RU1, where N^N = BPY [52]; complex RU2, where N^N = DOP [53]; heteroleptic Ru(II) complex RU3, where N^N = DPPN [54]; 21-carbon alkyl chain complexes RU4a and RU3b, where N^N = PHEN or TAP, respectively [55].

**Figure 6 biomedicines-10-00578-f006:**
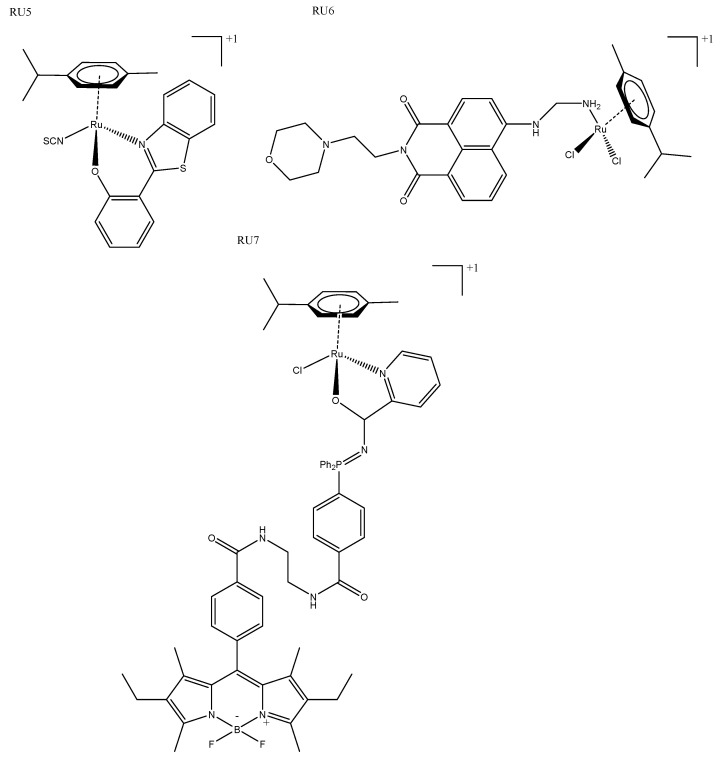
Ruthenium piano stool complexes RU5 [56], RU6 [57], and RU7 [58].

**Figure 7 biomedicines-10-00578-f007:**
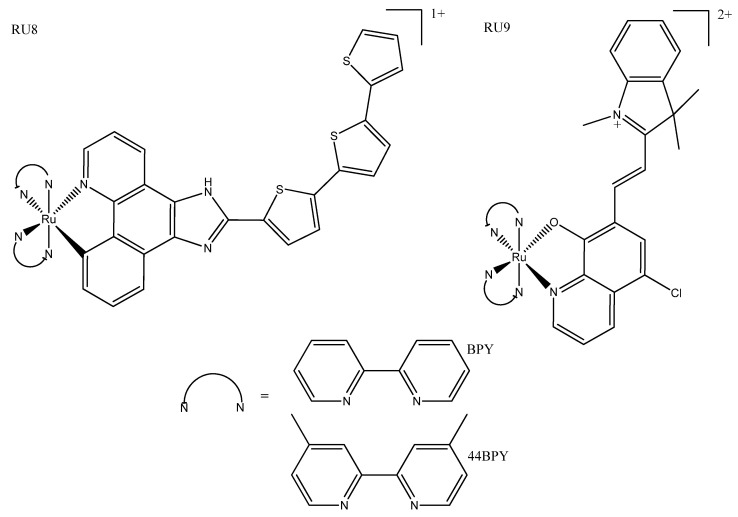
Cyclometallated ruthenium complexes RU8a and RU8b, where N^N = 44BPY or BPY, respectively [59,60]; N^O complex RU9, where N^N = BPY [61].

**Figure 8 biomedicines-10-00578-f008:**
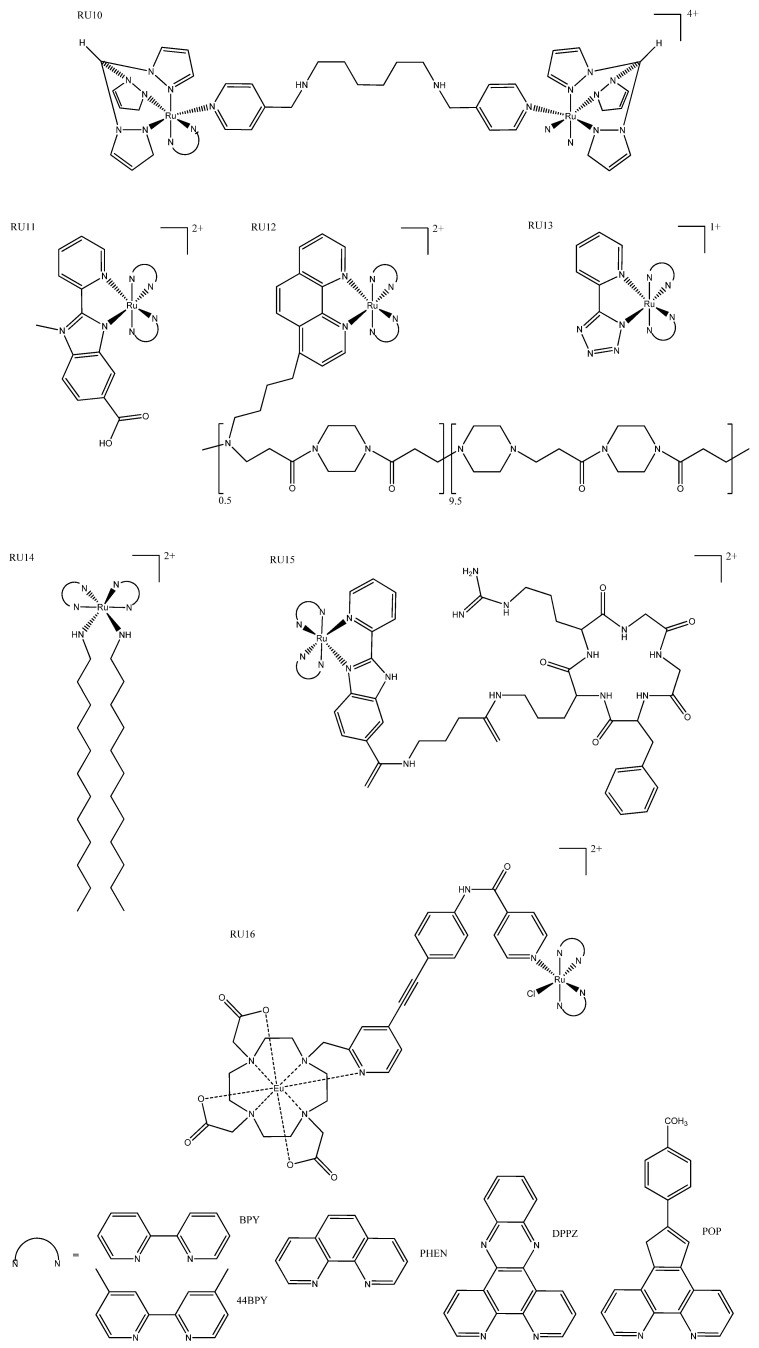
Other examples of ruthenium complexes: RU10, where N^N = DPPZ [62]; RU11, where N^N = BPY [63]; RU12, where N^N = PHEN [64]; RU13, where N^N = BPY [65]; RU14, where N^N = PHEN [66]; RU15, where N^N = POP [67]; and Eu-linked ruthenium complex RU16, where N^N = BPY [68].

**Figure 9 biomedicines-10-00578-f009:**
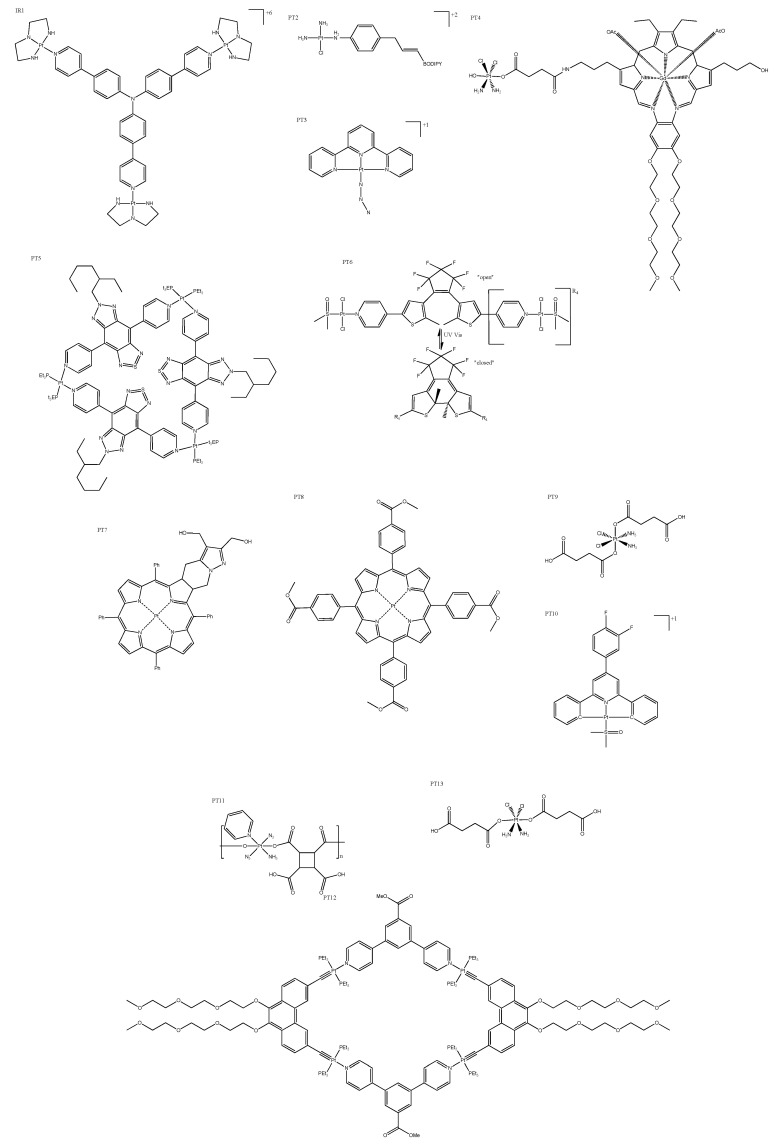
Photosensitizer platinum complexes: Y-shaped platinum complex PT1 [71]; square planar Pt(II) with weak photosensitizer, where BODIPY = boron–dipyrromethene, PT2 [72] and PT3 [73]; platinum–galadium complex PT4 [74]; triangular tri-platinum complex PT5 [75]; dual-platinum center Pt(II) diethienylcyclopentene PT6 in “open” and “closed” positions [76], PT7 [77], and PT8 [78]; platinum complexes (used to decorate a nanoparticle) PT9 [79] and PT10, [80]; platinum polymers PT11 [81] and PT12 [82]; and platinum complex (used as part of a nanoparticle) PT13 [83].

**Figure 10 biomedicines-10-00578-f010:**
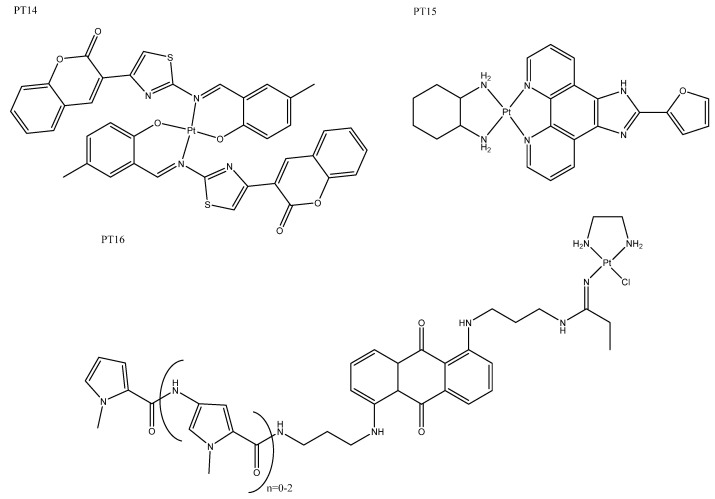
Platinum complexes with fluorescent moieties: PT13 [84]; PT15 Pt(II) imidazole phenanthroline complex [86]; and PT16 polyamide conjugated complex, where *n* = 2 [87].

**Figure 11 biomedicines-10-00578-f011:**
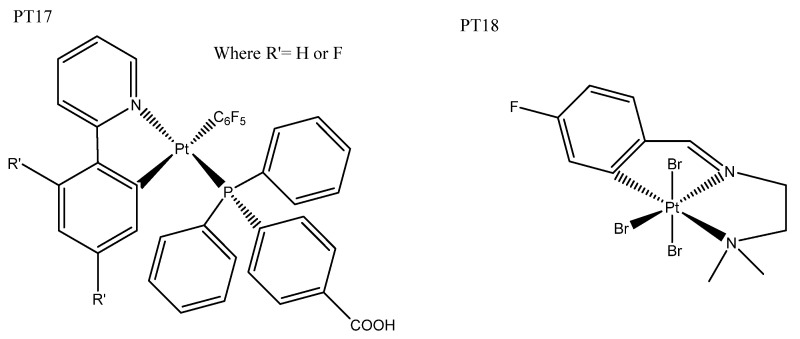
Cyclometallated platinum complexes PT17a, where R’ = H; PT17b, where R’ = F [88]; and PT18 [89].

**Figure 12 biomedicines-10-00578-f012:**
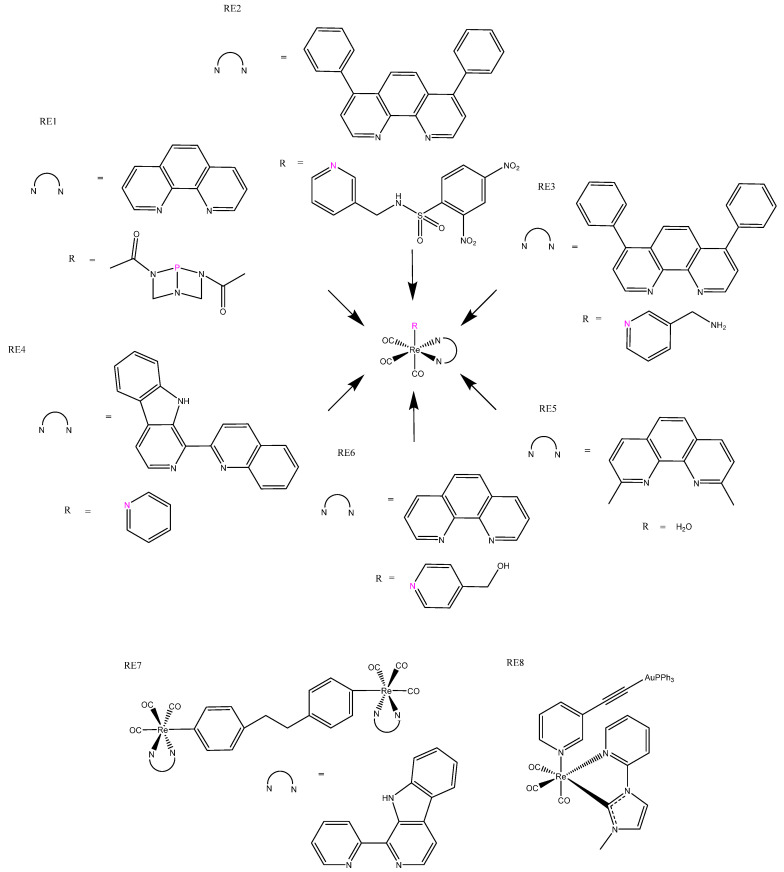
Rhenium(I) luminescent complexes: phosphine ligated complex RE1 [93], GHS-responsive bio-sensing complex RE2 [94], mitochondrial-targeted theragnostic complex RE3 [96], β-carboline rhenium complex RE4 [97], rhenium(I) aqua complex RE5 [98], apoptosis-ferroptosis dual-induction complex RE6 [99], heterodinuclear Re(I) β-carboline complexes RE7 [100], and gold(II)/rhenium(I) heterodinuclear complex RE6 [102].

**Figure 13 biomedicines-10-00578-f013:**
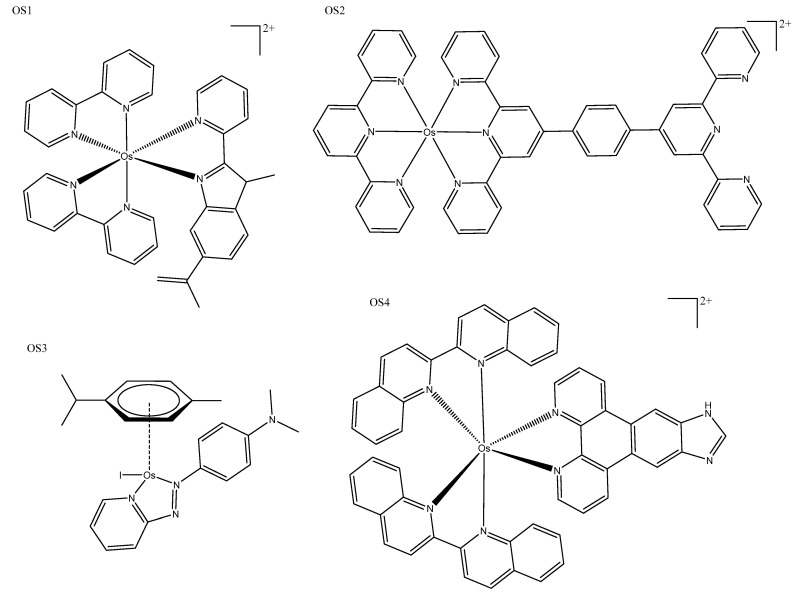
Osmium complex OS1 [104], tridentate terpyridine complex OS2 [105], half-sandwich osmium(II) complex OS3 [106], and panchromatic osmium(IV) complex OS4 [107].

**Figure 14 biomedicines-10-00578-f014:**
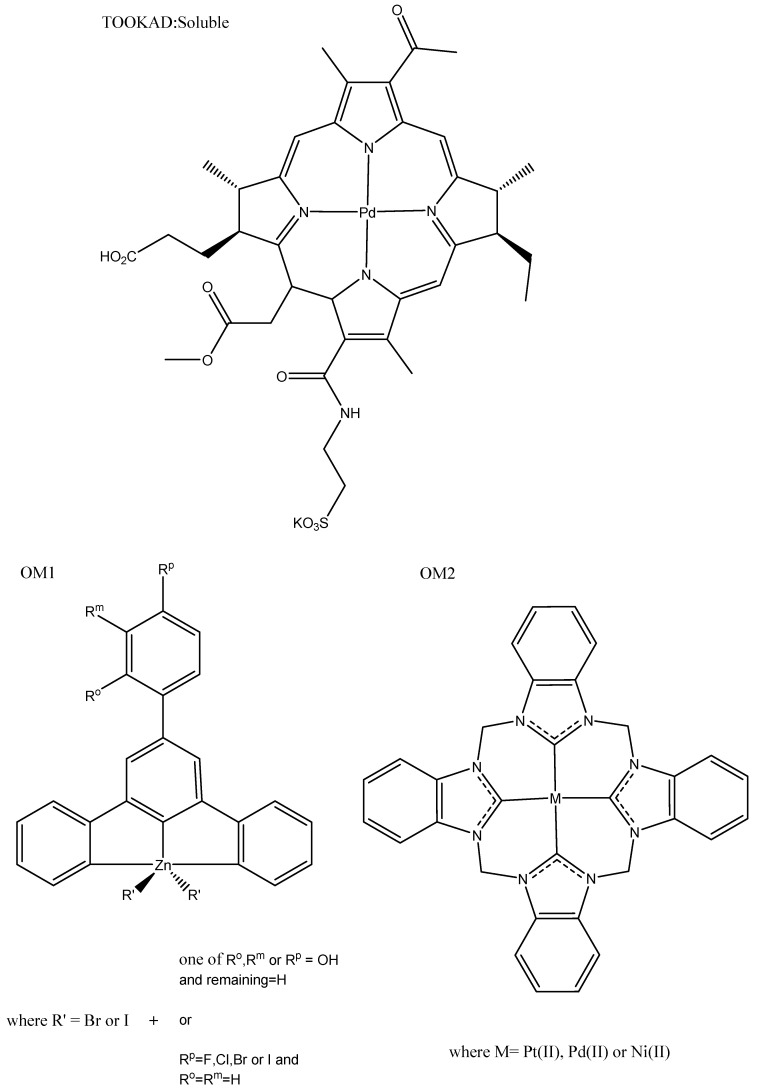
Examples of other transition metal fluorescent anticancer complexes [84,108,112,113].

**Table 1 biomedicines-10-00578-t001:** A summary of IC_50_ values, luminescent properties, and special features of iridium complexes: half sandwich complex IR13, where N^N = L1 [34]; half sandwich complex IR6, where N^N = L2 [24]; cyclometallated complexes IR2 [19], IR15 [37], IR16 [38], IR5 [39], IR21 [46]; dinuclear iridium(III) complex IR22 [47]; TRAIL-mimicking peptide complex IR23 [48]; and micelle multi-action prodrug IR24 [49].

Ir Complex	HeLa IC_50_		A549 IC_50_		Luminescence		Special Features
	Dark	Light (Irradiation nm)	Dark	Light (Irradiation nm)	λ_Em_ (nm)	λ_Ex_ (nm)	
IR2	2.1 ± 0.2		1.7 ± 0.1		630–690	405	Could cause concentration-dependent cell cycle arrest, pro-death autophagy, and caspase-dependent apoptosis in A549 cells.
IR6	11.3 ± 0.1		15.6 ± 1.2				Could induce cancer cell death in a variety of ways and had a good antimetastatic ability to cancer cells.
IR13	3.6 ± 0.8		2.6 ± 0.3		500–600	488	Fluorescence could be used for cellular imaging under both acidic and neutral conditions.
IR15	>100	0.21 ± 0.01 (425)	>100	0.31 ± 0.02 (425)	670	405	Had enhanced emission in lysosomes and could inhibit several key cancerous events, including cell migration, invasion, colony formation, and in vivo angiogenesis.
IR16			4.34 ± 0.01		591	488	Potential anticancer agents with dual functions, including metastasis-inhibition and lysosomal damage.
IR17			62.3 ± 2.6	1.1 ± 0.3 (465)	580–630	563	Photo-selective and cancer-selective in cell and spheroids.
IR21	11.5 ± 0.2	0.044 ± 0.006 (420)			557	381	The first iridium complexes to induce cancer cell death by inhibition of translation targeting the endoplasmic reticulum.
IR22	49.9 ± 0.1 *	0.75 ± 0.01 (visible)			608	405	Nanomolar photocytotoxicity and visible PI > 280.
IR23					506	366	The first example of an artificial TRAIL mimic that induced apoptosis-like cell death.
IR24					460–510	400	Folate targeted multi-action micelle, activated by GHS and irradiation at 400nm.

* IC_50_ values in SK-MEL-28 cells, as neither HeLa nor A549 have been published.

**Table 2 biomedicines-10-00578-t002:** A summary of the IC_50_ values, luminescent properties, and special features of ruthenium(II) complexes: TDL1433, where N^N = 44BPY [50]; similar complex RU1, where N^N = BPY [52]; heteroleptic Ru(II) complex RU3 [54]; 21-carbon alkyl chain complexes RU4a, where N^N = PHEN, and RU4b, where N^N = TAP [55]; piano stool complexes RU5 [56] and RU6 [57]; cyclometallated complexes RU8a, where N^N = 44BPY [59], and RU8b, where N^N = BPY [60]; and Eu linked ruthenium complex RU16 [68].

Ru Complex	HeLa IC_50_		SK-MEL-28 IC_50_		Luminescence		Special Features
	Dark	Light (Irradiation nm)	Dark	Light (Irradiation nm)	λ_Em_ (nm)	λ_Ex_ (nm)	
TDL1433			137 ± 3	1.9 ± 0.1 (400-700)		525	Optimized clinical procedure and completion of human clinical trials.
RU1	36.5 ± 3.0	3.1 ± 0.6 (420)			620	420	Photosensitizers for one- and two-photon PDT.
RU3			123 ± 3.62	3.77 ± 0.18	557/640	413	Activated at multiple wavelengths; tracking possible both before and after photo treatment.
RU4a	13 ± 2	0.47 ± 0.01 **			614	440	Shows how modification of ancillary ligand and lipophilicity enhances therapeutic effect.
RU4b	11 ± 3	2 ± 1 **			643	418	
RU6	83.1 ± 6.2 *	34.1 ± 2.4 (460)			500	355	Slight variations in structure led to phototoxic or other photoactivated complexes.
RU6	31.3 ± 4.5	11.5 ± 2.5 (488)			534	440	Tagged with napthalamide derivative to target DNA.
RU8a			>300	12.0 ± 0.4 (633)	805	540	More lipophilic and absorption; more red-shifted than their non-cyclometallated counterparts.
RU8b			>300	16.6 ± 1.53 (625)	728	550	
RU16	277.0 ± 7.1	32.5 ± 8.2 (488)			570-750	350	The linker could be irradiated at different wavelengths for different functionalization (prodrug activation or fluorescence).

* Using A549, as neither HeLa nor SK-MEL-28 were tested; ** no wavelength specified—“18 J cm^−2^ of light”.

**Table 3 biomedicines-10-00578-t003:** A summary of the IC_50_ values, luminescent properties, and special features of platinum-based luminescent anticancer complexes. Cyclometallated complexes PT17a, where R’ = H, and PT17b, where R’ = F [88]; Pt(II) imidazole phenanthroline complex PT15 [86]; platinum-decorated nanoparticle PT9 [79]; dual-platinum-center platinum(II) diethienylcyclopentene PT6 [76]; square planar Pt(II) with weak photosensitizer PT2 [72]; and polyamide conjugated complex PT16 [87].

Pt Complex	IC_50_ (μM)		Luminescence		Special Features
	Dark (Cell Line)	Light (Irradiation nm)	λ_Em(max)_ (nm)	λ_Ex(max)_ (nm)	
PT2	>50 (HeLa)	7.4 ± 0.3(635)	550–650	566	Positively charged Pt(II) center not only provided the cell membrane with an anchoring ability, but also made the complex a mild photosensitizer.
PT6	>75 (A375)	3.1 ± 0.2(365)	470	300	Light-mediated conversion from open to closed.
PT9	None (HeLa)	“Increased” (NIR) *			Negatively charged in normal physiological conditions, and converted to positive charge in acidic tumor extracellular microenvironments.
PT15			410, 470	300	DNA groove via hydrogenic or hydrophobic interaction.
PT16	16.2 ± 0.3 (MDA-MB-231)		615	525	Spheroid penetration.
PT17a	20.29 ± 2.10 (HeLa)		400	475	Shows potential as an anti-tubulin agent, and also provides useful information for our understanding of the mechanism responsible for the cytotoxic activity caused by these cyclometallated complexes.
PT17b	12.45 ± 2.50 (HeLa)		388	458	Selective generation of oxidative stress in cancer cells over noncancerous cells.

* No table of IC_50_ values nor a precise irradiation wavelength were provided.

**Table 4 biomedicines-10-00578-t004:** A summary of the IC_50_ values, luminescent properties and special features of rhenium(I) luminescent complexes: phosphine ligated complex RE1 [93], GHS-responsive bio-sensing complex RE2 [94], mitochondrial-targeted theragnostic complex RE3 [96], rhenium(I) aqua complex RE5 [98], heterodinuclear Re(I) β-carboline complex RE7 [100], and gold(II)/rhenium(I) heterodinuclear complex RE8 [102].

Re Complex	HeLa IC_50_ (μM)		A549 IC_50_ (μM)		Luminescence		Special Features
	Dark	Light (Irradiation nm)	Dark	Light (Irradiation nm)	λ_Em_ (nm)	λ_Ex_ (nm)	
RE1	>200	5.9 ± 1.4 (365)			516	~360	The phosphine ligand regulated luminescence, which strongly correlated to cytotoxicity.
RE2	6.50 ± 2.0				541	405	Showed significant emission enhancement when in the presence of GHS.
RE3	0.52 ± 0.07		3.4 ± 0.6		540	405	Effective repression of mitochondrial metabolism; had O_2_-sensitive phosphorescent lifetimes
RE5			6.7 ± 4.9		560–590		Induced caspase-independent cell death accompanied by cytoplasmic vacuolization.
RE7	4.0 ± 0.6		15.8 ± 10	0.26 ± 0.04 (425)	~550	405	Homodinuclear Re(I) for chemo-photodynamic therapy.
RE8			12.18 ± 1.19	4.48 ± 0.71 (405)	460-514	414	Heterodinuclear Au(II)–Re(I) complex; gold brought additional bioactivity.

**Table 5 biomedicines-10-00578-t005:** A summary of the IC_50_ values, luminescent properties, and special features of osmium complexes: tridentate terpyridine complex OS2 [105], half-sandwich osmium(II) complex OS3 [106], and, panchromatic osmium(IV) complex OS4 [107].

Os Complex	IC_50_ (μM)		Luminescence		Special Features
	Dark (Cell Line)	Light (Irradiation nm)	λEm(max) (nm)	λEx(max) (nm)	
OS2	>100 (Hep-G2)	1.23 ± 0.12 (465) 4.05 ± 0.05 (633)	~570–770	488	Exhibited high phototoxicities against cancer cells upon both 465 and 633 nm light irradiation.
OS3	1.1 ± 0.2(A549)				Selective generation of oxidative stress in cancer cells over noncancerous cells.
OS4	550 ± 49 (U87)	45 ± 5 (NIR)	940	200–900	These photosensitizers were panchromatic (i.e., black absorbers), activatable from 200 to 900 nm.

## Data Availability

Not applicable.

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
