# Peer review of "Photoactive and Luminescent Transition Metal Complexes as Anticancer Agents: A Guiding Light in the Search for New and Improved Cancer Treatments"

_biomedicines, 2022, doi:10.3390/biomedicines10030578_

Round 1

Reviewer 1 Report

In the review entitled “Luminescent transition metals as anticancer agents: A guiding light in the search for new and improved cancer treatments” the authors offered a panorama of the recent advances in the design of luminescent metal complexes for cancer treatment.

The topic is relevant for the scientific audience of Biomedicines and collect the most recent paper published in this regard. The structure of the review has been designed around the discussion of the most representative transition metal complexes of iridium(III), rutheniu(II), platinum(II), rhenium(I) and osmium(II). However, I have found many inaccuracies for instance in the text and also in the figures.

For example, iridium(III) complexes have been discussed first, but in the section 2.1 iridium(II) is cited again and again, whereas in section 2.3 both iridium(II) and iridium(III) are reported.

In section 3.2 the ruthenium complex RU6 is not a piano stool complex, RU5 should be cited instead; likewise, in the section 3.3 the ruthenium complex RU7a/b has been mentioned, but there is no correspondence with Figure 3, RU6a/b should be cited instead.

In many cases the structures of the complexes reported in the figures are not correct and do not report the net charge of the complexe.

Finally, the too many typos and the English language used make the reading very hard

For this reason I reject the review in this form; but, considering the relevance of the topic, I encourage the authors to carry out an extensive editing process and re-submit the review in a new form.

Author Response

Reviewer 1 comments

Our response

In the review entitled “Luminescent transition metals as anticancer agents: A guiding light in the search for new and improved cancer treatments” the authors offered a panorama of the recent advances in the design of luminescent metal complexes for cancer treatment.

The topic is relevant for the scientific audience of Biomedicines and collect the most recent paper published in this regard. The structure of the review has been designed around the discussion of the most representative transition metal complexes of iridium(III), ruthenium(II), platinum(II), rhenium(I) and osmium(II). However, I have found many inaccuracies for instance in the text and also in the figures.

 We thank the reviewer for their comments, we have incorporated all the advice given and are satisfied that we now have a much improved publication.

For example, iridium(III) complexes have been discussed first, but in the section 2.1 iridium(II) is cited again and again, whereas in section 2.3 both iridium(II) and iridium(III) are reported.

All instances of this typo has been resolved.

In section 3.2 the ruthenium complex RU6 is not a piano stool complex, RU5 should be cited instead; likewise, in the section 3.3 the ruthenium complex RU7a/b has been mentioned, but there is no correspondence with Figure 3, RU6a/b should be cited instead. 

Thank you, these errors have been resolved and double checked, as many of the complexes have been renumbered.

In many cases the structures of the complexes reported in the figures are not correct and do not report the net charge of the complexes.

Structures have been reviewed and missing charges added.

Finally, the too many typos and the English language used make the reading very hard

We have corrected many grammatical and spelling errors to align with UK English standards.

For this reason I reject the review in this form; but, considering the relevance of the topic, I encourage the authors to carry out an extensive editing process and re-submit the review in a new form.

We thank the reviewer for their comments and encouragement to resubmit, we believe we now have an improved manuscript thanks to the reviewers advice.

Reviewer 2 Report

This manuscript by McGhie and Aldrich-Wright reviews the recent literature about transition metal complexes with luminescent properties that can be used as anticancer agents. Efforts have been made on looking at how discrete structural changes influence the chemical and luminescent properties. The authors have not only focused on therapeutic agents but also on diagnostic agents and theranostic agents. Overallm this is a well-written, well-presented review on this topic. It is a useful reference that will be used by drug discovery researchers interested in luminescent metal complexes for biomedical applications. I recommend that the paper to be accepted for publication once the few minor erros are corrected (please see the file attached).

Author Response

Reviewer 2 comments

Our response

This manuscript by McGhie and Aldrich-Wright reviews the recent literature about transition metal complexes with luminescent properties that can be used as anticancer agents. Efforts have been made on looking at how discrete structural changes influence the chemical and luminescent properties. The authors have not only focused on therapeutic agents but also on diagnostic agents and theranostic agents. Overall this is a well-written, well-presented review on this topic. It is a useful reference that will be used by drug discovery researchers interested in luminescent metal complexes for biomedical applications. I recommend that the paper to be accepted for publication once the few minor errors are corrected (please see the file attached).

We thank the reviewer for their kind comments and recommendations.

We have made all the changed recommended by this reviewer in the PDF they submitted, including adding an additional sentence and reference to explain the importance of monitoring mitochondrial viscosity.

Reviewer 3 Report

The authors reviewed the recent progress of metal compounds, especially those with luminescent properties, as novel anticancer agents. This review article is comprehensive and critical, and can arouse broad interest in the field of bioinorganic chemistry, photochemistry, phototherapy and etc. I think this review article is suitable for publication on Biomedicines after some modifications listed as below.

  1. The title “Luminescent transition metals...” was a bit inconsistent with the content, which also contains many non-emissive metal compounds. Moreover, it is better to say “transition metal compounds” not “transition metals”.
  2. The authors need to specify the chemical structures with the corresponding names, especially for the ones have different ancillary ligands. Now it is a little difficult to find the right strucutres of IR1a, IR1b, RU3a, RU3b and etc.
  3. Only a few structures were displayed in the figures, which was not friendly to the readers.

Author Response

Reviewer 3 comments

Our response

The authors reviewed the recent progress of metal compounds, especially those with luminescent properties, as novel anticancer agents. This review article is comprehensive and critical, and can arouse broad interest in the field of bioinorganic chemistry, photochemistry, phototherapy and etc. I think this review article is suitable for publication on Biomedicines after some modifications listed as below.

We thank the reviewer for their kind words and advice; we have gladly incorporated all of their advice.

The title “Luminescent transition metals...” was a bit inconsistent with the content, which also contains many non-emissive metal compounds. Moreover, it is better to say “transition metal compounds” not “transition metals”.

The title has been adjusted to read

”Photoactive and luminescent transition metal complexes as anti-cancer agents: A guiding light in the search for new and improved cancer treatments”

Thank you for the correction

The authors need to specify the chemical structures with the corresponding names, especially for the ones have different ancillary ligands. Now it is a little difficult to find the right structures of IR1a, IR1b, RU3a, RU3b and etc

We have made significant changes to the figures in the manuscript thanks to these reviewers’ comments.

We have added the structure of all complexes mentioned and made sure that all structures are easily discernible when several variations are shown (i.e. When there is an “a” and “b” variation.

Round 2

Reviewer 1 Report

All the points raised in the previous report have been addressed and the paper has been edited accordingly.

In this form the paper is suitable for the publication in Biomedicines